# UNIMATE: A Unified Model for Mechanical Metamaterial Generation, Property Prediction, and Condition Confirmation

**Wangzhi Zhan** [1]  **Jianpeng Chen** [1]  **Dongqi Fu** [2]  **Dawei Zhou** [1]

## Abstract

Metamaterials are artificial materials that are designed to meet unseen properties in nature, such as ultra-stiffness and negative materials indices. In mechanical metamaterial design, three key modalities are typically involved, i.e., ***3D topology***, ***density condition***, and ***mechanical property***. Real-world complex application scenarios place the demanding requirements on machine learning models to consider all three modalities together. However, a comprehensive literature review indicates that most existing works only consider two modalities, e.g., predicting mechanical properties given the 3D topology or generating 3D topology given the required properties. Therefore, there is still a significant gap for the state-of-the-art machine learning models capturing the whole. Hence, we propose a unified model named **UNIMATE**, which consists of a modality alignment module and a synergetic diffusion generation module. Experiments indicate that UNIMATE outperforms the other baseline models in topology generation task, property prediction task, and condition confirmation task by up to 80.2%, 5.1%, and 50.2%, respectively. We open-source our proposed UNIMATE model and corresponding results at https://github.com/wzhan24/UniMate.

## 1. Introduction

Metamaterials are synthetic materials with unique properties, which are typically rarely observed in natural materials (Engheta & Ziolkowski, 2006). To be specific, mechanical metamaterials are a specific class of metamaterials designed to achieve unusual mechanical properties through

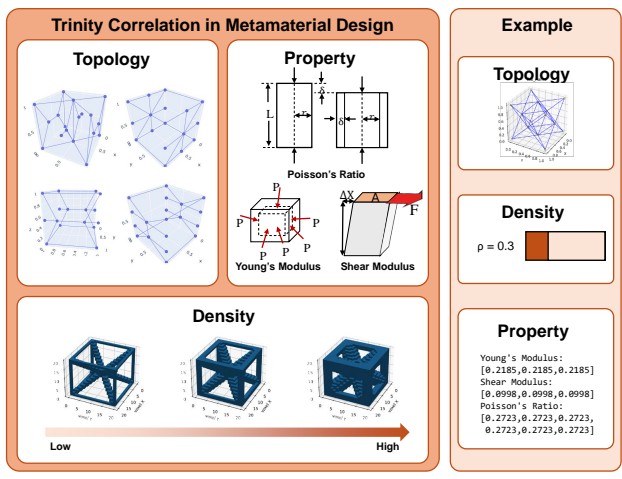

*Figure 1.* Trinity Representation of Metamaterials.

their engineered structures rather than the chemical composition. These materials are revolutionizing various fields, offering novel solutions for challenges in engineering, manufacturing, and materials science (Barchiesi et al., 2019). Metamaterials can have many possible important merits that classic materials do not have, like negative Poisson's ratio (Wang et al., 2020a) (favored for soft device), energy absorption behavior (Yuan et al., 2019) (favored for cushioning device), tunability (Sheng & Varadan, 2007) (favored for device with customized features), and extreme strength-to-weight ratio (Wang et al., 2020b) (favored for ultra-light and strong device). With the unique and superior merits of metamaterials, they have become an optimal choice in many engineering fields like energy storage, biomedical, acoustics, photonics, and thermal management (Surjadi et al., 2019).

Metamaterial design has a higher degree of freedom compared to traditional material design due to the possibility of manipulating and engineering the properties at the sub-wavelength level. In traditional material design, owing to natural laws, only a small amount of components can be tailored, like the percentage of each element, the phase of materials, etc. However, metamaterials can have artificially programmed structures as long as certain manufacturing requirements (e.g., periodicity) are satisfied (Sinha & Mukhopadhyay, 2023). Therefore, the design tasks are mul-

---

[1]Department of Computer Science, Virginia Polytechnic Institute and State University, Virginia, US [2]Meta AI. Correspondence to: Dawei Zhou <zhoud@vt.edu>.

*Proceedings of the $42^{nd}$ International Conference on Machine Learning*, Vancouver, Canada. PMLR 267, 2025. Copyright 2025 by the author(s).

*Table 1.* Comprehensive Review of Model's Ability for Metamaterial

| MODEL | TOPOLOGY GENERATION | PROPERTY PREDICTION | CONDITION CONFIRMATION |
| --- | --- | --- | --- |
| SCHNET (SCHÜTT ET AL., 2017) | ☹ | ☺ | 😐 |
| FTCP (REN ET AL., 2020) | ☺ | ☹ | ☹ |
| COND-DFC-VAE (COURT ET AL., 2020) | ☺ | ☺ | 😐 |
| PAINN (SCHÜTT ET AL., 2021) | ☹ | ☺ | 😐 |
| CDVAE (XIE ET AL., 2022) | ☺ | ☹ | ☹ |
| CG-SCHNET (GEBAUER ET AL., 2022) | ☺ | ☹ | ☹ |
| MCGCNN (MEYER ET AL., 2022) | ☹ | ☺ | 😐 |
| MACE (BATATIA ET AL., 2022) | ☹ | ☺ | 😐 |
| EQUIFORMER (LIAO & SMIDT, 2022) | ☹ | ☺ | 😐 |
| EDM (HOOGEBOOM ET AL., 2022) | ☺ | ☹ | ☹ |
| SPHERENET (LIU ET AL., 2022) | ☹ | ☺ | 😐 |
| DIFFCSP (JIAO ET AL., 2023) | ☺ | ☹ | ☹ |
| UNITRUSS (ZHENG ET AL., 2023A) | 😐 | ☺ | 😐 |
| VISNET (WANG ET AL., 2024) | ☹ | ☺ | 😐 |
| SYMAT (LUO ET AL., 2024B) | ☺ | ☹ | ☹ |
| EQUICSP (LIN ET AL., 2024) | ☺ | ☹ | ☹ |
| COND-CDVAE (LUO ET AL., 2024A) | ☺ | ☹ | ☹ |
| MACE+VE (GREGA ET AL., 2024) | ☹ | ☺ | 😐 |
| COMFORMER (YAN ET AL., 2024) | ☹ | ☺ | 😐 |
| UNIMATE (OURS) | ☺ | ☺ | ☺ |

tifaceted (Zheng et al., 2023a; Bastek et al., 2022b; Maurizi et al., 2022), e.g., giving part of the structure and generating the rest according to some desired property, giving part of the properties and generating a suitable structure and the rest of the property, giving the structures and predicting the possible properties, and giving the structures and properties and selecting the optimal conditions during the generation. Mastering all the above tasks is challenging and requires comprehensive knowledge and consideration of all possible aspects. To the best of our knowledge, there is no existing AI work that models all metamaterial design aspects together, as shown in Table 1, and the detailed review and why we need the comprehensive modeling are expressed below.

Formally, three key aspects (i.e., data modalities) compose the entire mechanical metamaterial design or generation, as shown in Figure 1, i.e., ***3D topology***, ***density condition***, and ***mechanical property*** (Yu et al., 2018). Knowing any two of these three aspects, we expect to derive the other one. For example, given the restrained density condition and the target property (e.g., stiffness), in this paper, we study how to empower generative models to design an effective structural organization among many possible 3D topologies, which task can be named as ***Topology Generation***, the other two tasks are named ***Condition Confirmation*** and ***Property Prediction***, their mathematical illustrations are expressed in Section 2.2.

The mechanical metamaterial community is eager for the comprehensive modeling method to address existing challenges: **(C1) Data Complexity**: mechanical metamaterial design involves three modalities with different formats and distributions. Hence it is difficult to utilize all such information; **(C2) Task Diversity**: given the complexity of data, any part of the data can be missing and requires design effort. Therefore there can be various design tasks; **(C3) Lack of Benchmark**: there is a lack of suitable benchmark that covers the diverse tasks, including a dataset providing varied topologies, density conditions and properties, and suitable metrics to evaluate the model performance. However, as shown in Table 1, existing works on material design are not satisfied, models like FTCP (Ren et al., 2020), SyMat (Luo et al., 2024b), EquiCSP (Lin et al., 2024) are solely targeted for topology generation. One relatively "versatile" model is Cond-DFC-VAE (Court et al., 2020), which can perform both structure generation and property prediction. However, Cond-DFC-VAE can only take structural information as input when performing property prediction tasks, such that when adopting it to the condition selection, the input to be handled is biased, i.e., only topological information is taken.

Motivated by the above analysis, we aim to propose a unified model that is versatile enough and can tackle three data modalities together in mechanical metamaterial design. Our proposed UNIMATE consists of two novel components, a modality alignment module and a synergetic generation module. The modality alignment module compresses the three different modalities into a shared latent space and uses tripartite optimal transport to align the latent tokens from each modality. The synergetic generation module receives all the latent tokens (some are unknown) and completes the unknown tokens through a score-based diffusion model.

Finally the denoised tokens from the diffusion model are reverted by a decoder(s) into the raw design space. In principle, the **modality alignment module** helps to narrow the large gap among the three modalities, which significantly alleviates challenge **C1**; the **synergetic generation module**, owing to its intrinsic flexible nature, is well-suited for address challenge **C2**. We propose a new dataset and new metrics, which help to solve challenge **C3**.

This work has the following main contributions:

- **Problem Formalization**: We propose and formalize a general design task for mechanical metamaterial design. The task defined in this work can be generalized to many specific important tasks, including topology generation, property prediction, and condition confirmation.

- **Unified Model**: We propose a unified model to address the general design task. This unified model can perform diverse kinds of tasks in mechanical metamaterial design.

- **Benchmarking and Experimentation**: We propose a suitable benchmark for the metamaterial design tasks and compare our model with six baselines. The result shows the superiority of our model.

## 2. Preliminary

Here, we briefly and formally introduce the background of metamaterial and relevant main tasks.

We use regular letters to denote scalars (e.g., $n$), boldface lowercase letters to denote vectors (e.g., $\mathbf{v_i}$), and boldface uppercase letters to denote matrices (e.g., $\mathbf{W_i}$), italic uppercase letters to denote sets (e.g., $\mathcal{L}$).

### 2.1. Background of Mechanical Metamaterial

As shown disentangled in Figure 2, mechanical metamaterials are basically substrate material accumulated according to synthetic spatial structures, i.e., lattice structures. To avoid complicated design, lattice structures are normally set to be periodic so that the entire body of metamaterial is made of a repeating smallest cell, called a unit cell. Therefore, the definition of the topology of metamaterial can be expressed as the topology of a unit cell (essentially a 3D graph) and three lattice vectors indicating how the unit cell is repeated. Given such topology information, we can customize how "densely" substrate material is accumulated along each edge (i.e., relative density[1] condition). When 3D topology and relative density conditions are determined, the mechanical properties of the metamaterial can be estimated, including

---

[1]In this paper, we use "relative density" and "density" interchangeably

Young's modulus, shear modulus, and Poisson's ratio (Wortman & Evans, 1965). Hence, we formally define mechanical metamaterial as follows.

**Metamaterial Trinity Representation** (MTR) represents a complete metamaterial representation, denoted as $\mathcal{M} = (\mathcal{T}, \rho, \boldsymbol{p})$, where $\mathcal{M}$ is a tuple of 3D topology $\mathcal{T}$, density condition $\rho$, and mechanical properties $\boldsymbol{p}$.

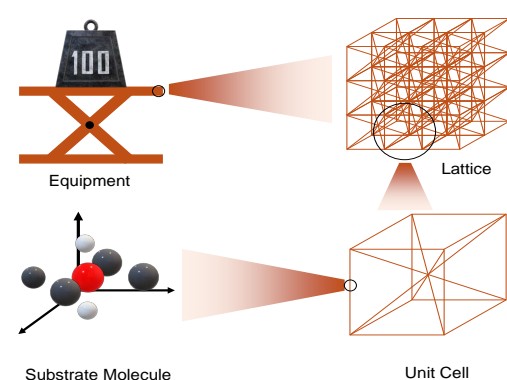

*Figure 2.* Hierarchical Composition of Mechanical Metamaterial.

**3D Topology** is represented by $\mathcal{T} = (\boldsymbol{L}, \boldsymbol{X}, \boldsymbol{A})$, where $\boldsymbol{L} = [\boldsymbol{l}_1, \boldsymbol{l}_2, \boldsymbol{l}_3]$ contains the three lattice vectors indicating the axes along which the unit cell is stacked; $\boldsymbol{X} \in \mathbb{R}^{n \times 3}$ is the 3D coordinates matrix of nodes, and $\boldsymbol{A} \in \mathbb{R}^{n \times n}$ is the adjacency matrix without self-loop, where $n$ denotes the node numbers, with $A_{i,j} = 1$ if $i$th node connects $j$th node.

**Relative Density Condition** is denoted as $\rho = \frac{\int_{\tilde{\Omega}} \mathrm{d}\omega}{\int_{\Omega} \mathrm{d}\omega}$, where $\Omega$ and $\tilde{\Omega}$ stands for the unit cell zone and solid zone within the unit cell, and $\mathrm{d}\omega$ is a small volume; in other words, density means the ratio of solid matter within the unit cell.

**Mechanical Properties** is denoted as $\boldsymbol{p}$, a vector indicating the mechanical attributes of materials, such as Young's modulus, shear modulus, Poisson's ratio (Wortman & Evans, 1965), etc.

The definition of MTR allows us to formalize the problem of designing a mechanical metamaterial as deriving a complete MTR.

### 2.2. Main Tasks

Given the background in Section 2.1, to derive a complete MTR, we focus on three basic tasks with different unknown components of the MTR concerned.

**Conditional Topology Generation**. Given restrained relative density $\rho$ and desired property $\boldsymbol{p}$, we aim to generate 3D topology $\mathcal{T}$ that meets certain geometric requirements (periodicity and symmetry) and matches the given property under the given density.

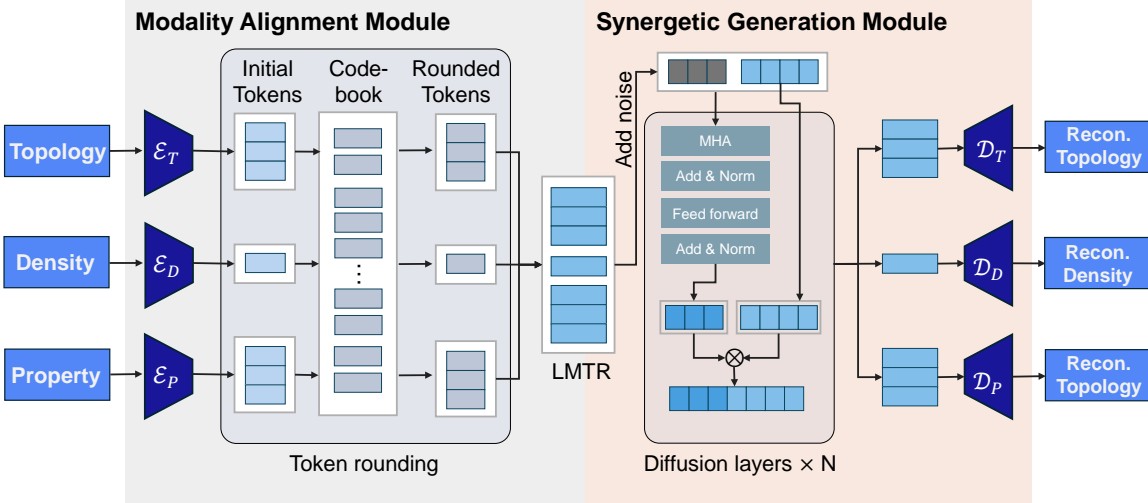

*Figure 3.* Pipeline Illustration of UNIMATE. Three modalities are fed into the model, crossing two modules, and generating reconstructed data for training. In the inference phase, the targeted modality data is generated from noise.

**Property Prediction**. Given the topology $\mathcal{T}$ and density condition $\rho$, we aim to predict the corresponding property $\boldsymbol{p}$, i.e., whether and how well a specific topology can perform under a particular density.

**Condition Confirmation**. Given topology $\mathcal{T}$ and property $\boldsymbol{p}$, we aim to confirm the corresponding density condition $\rho$. This task helps the researchers to know how light or heavy the designed material can be, with the given topology and given property. This task holds particular significance in designing materials that are light yet exhibit high stiffness, as well as in the development of lightweight cushioning materials.

## 3. Methodology

In this section, we introduce the details of the proposed UNIMATE model that is designed to handle diverse tasks in a unified framework for metamaterial design. We begin with an overview of the model in Section 3.1, which consists of two main modules, i.e., Metamaterial Modality Alignment in Section 3.2 and Metamaterial Synergetic Generation in Section 3.3. Section 3.4 explains the training and inference procedures of our model.

### 3.1. Overview

There are three key challenges in mechanical metamaterial design, i.e., data complexity, task diversity, and lack of benchmark. To demonstrate this, in this paper, we first prepare a benchmark to solve the third challenge together, and the details are given in Appendix A.

To be specific, for addressing the first two challenges, we propose a model with two specially designed modules, i.e.,

one for modality alignment and the other for a synergetic generation as shown in Figure 3. The modality alignment module maps the raw information into a shared latent space and aligns the latent tokens from each modality. The synergetic generation module takes the known latent tokens as context and generates the unknown tokens.

Next, we will elaborate on the technical details of metamaterial modality alignment and metamaterial synergetic generation.

### 3.2. Metamaterial Modality Alignment

In brief, the modality alignment module is designed to align the three modalities, thus facilitating the generation process and addressing the challenge C1.

Since continuous spaces are more difficult to align, inspired by inspired by VQ-VAE (Van Den Oord et al., 2017), we first map the raw data into a discrete latent space. Then, we use three pairs of encoders and decoders to map the data from raw format into the latent space. These tokens are then mapped into a shared discrete latent space by comparing with a codebook, which is essentially a set of prototype tokens $\mathcal{Z} = \{\boldsymbol{z}_i\}_{i=1}^{\kappa}$, where $\boldsymbol{z}_i \in \mathbb{R}^d$, $\kappa$ is the codebook size, and $d$ is the token dimension.

The three pairs of encoders $\mathcal{E}$ and decoders $\mathcal{D}$ corresponding to the three different modalities for topology $\mathcal{T}$, density $\rho$, and property $\boldsymbol{p}$ are denoted as $\{\mathcal{E}_\mathcal{T}, \mathcal{D}_\mathcal{T}\}$, $\{\mathcal{E}_\rho, \mathcal{D}_\rho\}$, and $\{\mathcal{E}_{\boldsymbol{p}}, \mathcal{D}_{\boldsymbol{p}}\}$. For example, $\mathcal{E}_\mathcal{T}$ is a graph convolutional network (GCN) defined as

$$\begin{aligned} \tilde{\boldsymbol{X}} &= \mathcal{E}_\mathcal{T}(\boldsymbol{X}, \boldsymbol{A}) = \text{GCN}^{i=1,\dots,L}(\boldsymbol{X}, \boldsymbol{A}), \\ \tilde{\boldsymbol{X}}^l &= \text{GCN}^l(\tilde{\boldsymbol{X}}^{l-1}, \boldsymbol{A}) = \boldsymbol{A}\sigma(\tilde{\boldsymbol{X}}^{l-1}, \boldsymbol{A})\boldsymbol{W}^l, \end{aligned} \tag{1}$$

where $\tilde{X}^0 = X$ and $1 \leq l \leq L$ denotes the number of layers. $W^l$ denotes the $l$th layer's parameters, $\sigma$ denotes Sigmoid activation function, and $\tilde{X}$ is the latent embedding consisting of latent tokens $\tilde{x}_i$.

After the latent tokens (e.g., $\tilde{x}_i$) obtained, We introduce a codebook to "round" the latent tokens, i.e., find the nearest prototype token in the codebook and substitute the latent tokens with the prototype token. The subscription "round" (e.g., $\tilde{X}_{\text{round}}$) below means rounded token(s).

Then, $\mathcal{D}_{\mathcal{T}}$ is introduced to decode the rounded tokens, consisting of several transformer layers with a final linear layer:

$$\hat{X} = \mathcal{D}_{\mathcal{T}}(\tilde{X}_{\text{round}}) = \text{TransLayer}^{i=1,\ldots,L}(\tilde{X}_{\text{round}})W_{\mathcal{T}}, \quad (2)$$

where each $\text{TransLayer}^l$ is a transformer layer using multi-head self-attention, and $W_{\mathcal{T}} \in \mathbb{R}^{d \times 3}$ is to calculate the vertex coordinates. Consequently, the reconstructed the adjacency matrix $\hat{A}$ can be obtained by

$$\hat{A}_{i,j} = \text{CosSim}(\hat{x}_i, \hat{x}_j), \quad (3)$$

where CosSim denotes cosine similarity.

In addition to coordinates $X$ and adjacency matrix $A$, to derive the MTR, the lattice vector $L$ can be calculated according to the method mentioned in Appendix A.2.

Similarly, $\mathcal{E}_{\rho}$ and $\mathcal{D}_{\rho}$ are each a multilayer perceptron (MLP) to process density $\rho$ and its latent representation $\tilde{\rho}$:

$$\tilde{\rho} = \mathcal{E}_{\rho}(\rho), \text{ and } \hat{\rho} = \mathcal{D}_{\rho}(\tilde{\rho}_{\text{round}}), \quad (4)$$

Also, $\mathcal{E}_p$ and $\mathcal{D}_p$ are two series of MLPs, each MLP for one dimension of $p$. The encoder maps the property $p$ into a series of tokens $\tilde{P}$, while the decoder does the reverse operation.

It should be noted that the output of the encoders, i.e., $\tilde{X}, \tilde{\rho}, \tilde{P}$, are not directly input into the decoders, but after "rounding" them into the nearest tokens in the codebook $\mathcal{Z}$. $\tilde{X}, \tilde{\rho}, \tilde{P}$ are concatenated into latent MTR (LMTR):

$$\tilde{M} = [\tilde{X}; \tilde{\rho}; \tilde{P}]. \quad (5)$$

$\tilde{M}$ can also be reformatted as a series of tokens: $\tilde{M} = [\tilde{m}_0, \tilde{m}_1, \ldots, \tilde{m}_h]$, where each $\tilde{m} \in \mathbb{R}^d$ is a latent token, $h$ is the LMTR token number, $d$ is the token dimension (also named latent dimension). For each token $m$ in $\tilde{M}$, the rounding process is performed as:

$$\tilde{m}_{\text{round},i} = z_j, \text{ where } j = \min_k \|\tilde{m}_i - z_k\|, \quad (6)$$

then, we get $[\tilde{X}_{\text{round}}; \tilde{\rho}_{\text{round}}; \tilde{P}_{\text{round}}]$.

The above three different modalities are being mapped into a shared discrete space comprising a series of tokens by the encoders and the codebook. A suitable method is needed

to align the three modalities. Optimal transport (Sinkhorn, 1967) (OT) provides a way to align two different distributions. To apply OT in our task, we generalize it into tripartite OT (TOT), which considers three modalities. The process of implementing TOT is to lower the tripartite Wasserstein distance (TWD) by optimizing the latent tokens. This optimization is then realized by optimizing the encoders mentioned in Section 3.1. The TWD, $d_{\text{w}}$, is defined as

$$d_{\text{w}} = \inf_{\lambda(\alpha,\beta,\gamma) \in \Lambda(\alpha,\beta,\gamma)} \mathbb{E}_{x,y,z \in \lambda(\alpha,\beta,\gamma)} [c(x,y,z)], \quad (7)$$

where $\alpha$, $\beta$ and $\gamma$ are the marginal distribution of $x$, $y$ and $z$, and $c(x, y, z)$ is the tripartite cost defined as

$$c(x,y,z) = \text{CosSim}(x,y) + \text{CosSim}(y,z) + \text{CosSim}(x,z). \quad (8)$$

Minimizing the $d_{\text{w}}$ in Equation (7) by optimizing the distribution of the three modalities $x$, $y$ and $z$ provides a way to align the three modalities. To achieve this, we propose a tripartite Sinkhorn algorithm (TSA), which is a generation of (Sinkhorn, 1967). The details can be found in Appendix B.2. Briefly speaking, the vanilla Sinkhorn algorithm (Sinkhorn, 1967) involves two distributions, such that to process the three modalities, we need to generalize the vanilla Sinkhorn algorithm into the tripartite Sinkhorn algorithm by generalizing the **transport plan** iteration process and adding the preprocessing (calculating the token frequency first, i.e., lines 3 to 10 in Alg. 1).

Alg. 1 gives the detailed implementation of our proposed TSA. Firstly, the marginal distribution of $\tilde{X}_{\text{round}}$, $\tilde{\rho}_{\text{round}}$ and $\tilde{P}_{\text{round}}$ is calculated with respect to the latent token series $\{z_i\}$ (lines 2 to 10 in Alg. 1). According to TSA, the transport plan (which is essentially a 3D tensor) can be expressed as $\text{TransPlan} = u \odot v \odot w \otimes M$, where $u$, $v$ and $w$ are three bases and $M$ is a damping matrix that is determined by the latent token series. Therefore, only $u$, $v$, and $w$ need to be calculated. Finally, lines 12 to 17 in Alg. 1 calculate the three bases in an iterative way.

With respect to the alignment operation, to optimize the model, the alignment loss can be defined as

$$\begin{aligned} L_{\text{align}} = \ &\alpha_{\mathcal{T}}(\|X - \hat{X}\| + \|\tilde{X} - \tilde{X}_{\text{round}}\| + \|A - \hat{A}\|) \\ &+ \alpha_{\rho}(\|\rho - \hat{\rho}\| + \|\tilde{\rho} - \tilde{\rho}_{\text{round}}\|) \\ &+ \alpha_p(\|p - \hat{p}\| + \|\tilde{P} - \tilde{P}_{\text{round}}\|) \\ &+ \alpha_{\text{w}} d_{\text{w}}. \end{aligned} \quad (9)$$

where $\alpha_{\mathcal{T}}$, $\alpha_{\rho}$, $\alpha_p$ and $\alpha_{\text{w}}$ are four hyperparameters.

After minimizing this alignment loss, we can obtain a 3D tensor that indicates the collaborative distribution of the three (latent) representations. Therefore, in the generation stage, instead of randomly choosing one starting point from pure noise, we can choose from the elements with higher probability as the starting point. This operation can decrease the discrepancy between the starting point and ending point

of the diffusion process. Hence, it will be simpler to generate proper outputs, which increase the robustness of the generation process.

## 3.3. Metamaterial Synergetic Generation

To address challenge C2, the synergetic generation module is proposed for diverse tasks during metamaterial design.

We implement the synergetic generation process as a partially frozen diffusion process, since a diffusion model (Song & Ermon, 2019) can take inputs with flexible shape if the backbone is a flexible model, while the frozen operation can provide context for the diffusion process.

The input to the synergetic generation module is the rounded LMTR $\tilde{M}_{\mathrm{round}}$, which is a series of tokens. Part of the input tokens are added with noise and are considered as unknown tokens. The indices of the known tokens and unknown tokens are denoted as $\mathcal{I}_{\mathrm{kn}}$ and $\mathcal{I}_{\mathrm{un}}$, respectively, with $\mathcal{I}_{\mathrm{kn}} \cup \mathcal{I}_{\mathrm{un}} = \mathcal{I} = \{0, 1, \ldots, h-1\}$, where $h$ is the number of all tokens. The known and unknown tokens can be denoted as $\tilde{M}_{\mathrm{round}}(\mathcal{I}_{\mathrm{kn}})$ and $\tilde{M}_{\mathrm{round}}(\mathcal{I}_{\mathrm{un}})$. The score-based diffusion process (Song & Ermon, 2019) can be expressed as a series of denoising steps, with each step being:

$$\tilde{M}_t = \tilde{M}_{t-1} + \frac{\alpha_i}{2}\phi(\tilde{M}_{t-1}) + \sqrt{\alpha_i}Z_t, \qquad (10)$$

where $t \in \{0, 1, \ldots, T\}$, $T$ is the time step number, $\alpha_i$ is a scalar declining with the diffusion process (we use the same setting of $\alpha_i$ as in (Song & Ermon, 2019)), $\phi$ is a transformer backbone, and $Z_t$ is Gaussian noise with the same shape as $\tilde{M}_t$. $\tilde{M}_0$ is set to be $\tilde{M}_{\mathrm{round}}$.

To keep the given context tokens intact, the transformer backbone performs **partially frozen diffusion**. In other words, for partially frozen diffusion, all the tokens go through each layer of the transformer and yield an output token series. Then, the formerly known tokens in the output are replaced with their initial values. Such partially frozen processing can be expressed formally as:

$$\phi(\tilde{M}, \mathcal{I}_{\mathrm{un}}) = [\phi(\tilde{M})(\mathcal{I}_{\mathrm{un}}); \tilde{M}(\mathcal{I}_{\mathrm{kn}})]. \qquad (11)$$

Then, Equation (10) becomes

$$\tilde{M}_t = \tilde{M}_{t-1} + \frac{\alpha_i}{2}\phi(\tilde{M}, \mathcal{I}_{\mathrm{un}}) + \sqrt{\alpha_i}Z_t. \qquad (12)$$

Owing to such a partially frozen operation and the intrinsic mechanism of the transformer, our model can take a series of tokens with arbitrary series length (except for memory limit's sake), while an arbitrary subset of the tokens can be set as unknown. The known tokens provide context information for other tokens, especially in the attention operation of the transformer. Therefore, we name such a

generation as synergetic generation. Finally, the generation loss is defined as

$$L_{\mathrm{gen}} = \|\tilde{M}_{\mathrm{round}} - \tilde{M}_T\|. \qquad (13)$$

## 3.4. Optimization and Inference

The training process is as follows: (1) the encoders project the raw MTR into latent space; (2) the codebook is used to round the latent tokens; (3) TOT is used to align different modalities; (4) the decoders are used to reconstruct the raw MTR; (5) add Gaussian noise to one random type of tokens (e.g., topology tokens) to obtained noised LMTR; (6) the noised LMTR is input into the diffusion model for denoising (i.e., the synergetic generation process). Note that steps 4 and 5 do not rely on each other, so they are two parallel operations. The training loss is

$$L_1 = \lambda_{\mathrm{align}}L_{\mathrm{align}} + \lambda_{\mathrm{gen}}L_{\mathrm{gen}}, \qquad (14)$$

which basically considers the MTR reconstruction error, the token rounding error, the modality align error and the error between generated tokens and initial latent tokens.

When using the model for testing or inference, it is assumed that part of the raw MTR is provided (although the model can also do unconditional generation to provide a random design). Then, the given part goes through the corresponding encoder(s), and some latent tokens are provided accordingly. Then, the unknown tokens are initialized with the conditional probability according to the transport plan to provide a high-quality initialization that is aligned with the input condition. The diffusion model then generates the completed $\tilde{M}_{\mathrm{round}}$, which then is reverted to the raw design space by the decoders.

# 4. Experiments

## 4.1. Experiment Setup

**Datasets**. Existing works lack a suitable dataset covering all three tasks in Section 2.2. To address this, we construct a new dataset based on (Lumpe & Stankovic, 2021) by selecting samples, assigning multiple density conditions to each topology, and computing mechanical properties via finite element simulation. Further details are in Appendix A.1.

**Baselines**. As detailed in Section 5, existing methods related to our work fall into generation-oriented and prediction-oriented models. For generation, we use CDVAE (Xie et al., 2022) and SyMat (Luo et al., 2024b) as baselines, both designed for periodic crystal structure generation. For prediction, we include Equiformer (Liao & Smidt, 2022), ViS-Net (Wang et al., 2024), MACE-ve (Grega et al., 2024), and uniTruss (Zheng et al., 2023a), which predict (meta)material properties from topology. Additionally, we adapt uniTruss for generation using its reconstruction capability.

*Table 2.* Effectiveness Comparison.

| MODEL | TOPO. GEN. TASK | | PROP. PRED. TASK | COND. CONFIRM. TASK |
| | $F_{\text{qua}}$ $(\times 10^{-2}, \downarrow)$ | $F_{\text{cond}}$ $(\times 10^{-2}, \downarrow)$ | $\text{NRMSE}_{\text{pp}}$ $(\times 10^{-2}, \downarrow)$ | $\text{NRMSE}_{\text{cc}}$ $(\times 10^{-2}, \downarrow)$ |
| --- | --- | --- | --- | --- |
| CDVAE (XIE ET AL., 2022) | 19.23 | 32.71 | N/A | N/A |
| EQUIFORMER (LIAO & SMIDT, 2022) | N/A | N/A | 5.31 | 38.05 |
| VISNET (WANG ET AL., 2024) | N/A | N/A | 3.12 | 10.43 |
| SYMAT (LUO ET AL., 2024B) | 16.94 | 33.37 | N/A | N/A |
| UNITRUSS (ZHENG ET AL., 2023A) | 19.43 | 33.77 | 2.71 | 8.89 |
| MACE+VE (GREGA ET AL., 2024) | N/A | N/A | 2.57 | 9.09 |
| UNIMATE (OURS) | **2.74** | **7.81** | **2.44** | **4.43** |

## 4.2. Effectiveness Analysis

To evaluate the effectiveness of our model, we compare its performance with the baseline models on the three tasks mentioned in Section 2.2. For topology generation, we consider the two generation-oriented baselines. For property prediction and condition confirmation, we consider the four prediction-oriented baselines. Existing models are not well suited for condition confirmation since they are not designed to input 3D topology and mechanical properties simultaneously. Therefore, we revise the three prediction-oriented models by forcing them to predict density for the condition confirmation task. The metrics we use to assess the generation result are $F_{\text{qua}}$ and $F_{\text{cond}}$. Specifically, $F_{\text{qua}}$ measures the topology quality, including symmetry and periodicity, while $F_{\text{cond}}$ measures how closely the result matches the ground truth topology. Details regarding these two metrics can be found in Appendix A.2. For the property prediction and condition confirmation tasks, we simply calculate the normalized root mean square error (NRMSE) between the model's output and the ground truth value.

As listed in Table 2, our model outperforms other baselines in all three tasks. In the topology generation task, property prediction task, and condition confirmation task, our model outperforms the second-best model by 80.2%, 5.1%, and 50.2%, respectively. This result verifies the superiority of our model in terms of effectiveness.

## 4.3. Time and Space Efficiency

To compare the time efficiency of our model with other models, we train each model on our dataset with varied batch sizes and record the average time required for processing each batch. The result is shown in Figure 4. According to Figure 4, the batch processing time for each model is roughly in linear relation to the batch size. In terms of the slope of each line in Figure 4, our model has a medium-level slope, implying a medium-level time efficiency. Additional results for other baselines can be found in Appendix C.1.

However, it should be noted that three lines in Figure 4 stop at smaller batch sizes. The reason is that these models trig-

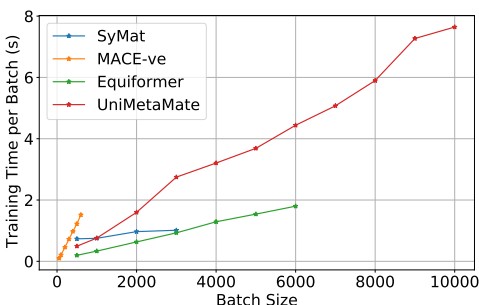

*Figure 4.* Comparison of Time Efficiency.

ger an out-of-memory error with the GPU near the stopping point of each line. Our model, on the other hand, triggers no error even after the batch size of 10000. Therefore, our model has a far higher space efficiency than other models considered.

## 4.4. Parameter Sensitivity

We also study the parameter sensitivity of our model by tuning the latent token dimension $d$ and number of tokens within the codebook $n$. The resulting generation metric $F_{\text{qua}}$ is shown in Figure 5, with other metrics in Appendix C.2. From Figure 5, it can be seen that our model performs better roughly with a larger latent token dimension and larger codebook size.

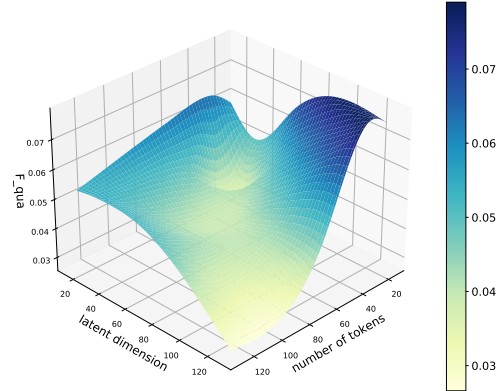

*Figure 5.* $F_{\text{qua}}$ under different parameters.

## 4.5. Ablation Study

Table 2 shows that our model performs well in capturing the cross-modality correlation. For the ablation study, 3 cases are used to figure out how much the diffusion generation module (with/without the partially frozen diffusion technique), the encoder-decoder compression module, and the alignment operation contribute to our model's correlation-capturing ability.

*Table 3.* Ablation Study.

|         | $F_{\text{qua}}(\downarrow)$ | $F_{\text{cond}}(\downarrow)$ | $\text{NRMSE}_{\text{pp}}(\downarrow)$ | $\text{NRMSE}_{\text{dp}}(\downarrow)$ |
|---------|--------|--------|--------|--------|
| CASE 1  | 0.0531 | 0.1136 | 0.0472 | 0.0757 |
| CASE 2  | 0.0295 | 0.0812 | 0.0263 | 0.0510 |
| CASE 3  | 0.0405 | 0.0813 | 0.0322 | 0.0423 |
| CASE 4  | 0.0274 | 0.0781 | 0.0244 | 0.0443 |

**Ablation case 1: (diffusion model alone)** Instead of generation in the LMTR space, the diffusion model can also be used to directly generate samples in the raw MTR space (although a linear layer is needed to bridge the gap between the input space dimension and transformer layers' processing dimension).

**Ablation case 2: (mapping into latent space + diffusion model)** In this case, we still use the encoder-decoder module to compress the original MTR into latent MTR and then implement the diffusion module without alignment operation.

**Ablation case 3: (vanilla diffusion)** In this case, we simply remove the partially frozen diffusion technique from the full model. Specifically, this means that all tokens, including the context tokens that should not be changed, are changed through each layer in the diffusion backbone.

**Ablation case 4: (full model)** This case corresponds to the full model, including design space unification, latent space alignment, and synergetic generation.

Table 3 lists the result of the ablation study. From Table 3 it can be seen that the unification and the alignment operation are both beneficial to our model's performance. The unification process boosts the performance by 37.5% in average, and the TOT alignment provides 7.8% boost upon case 2. By comparing case 3 and case 4 it can be seen that the partially frozen diffusion technique provides a 13.9% performance boost in average, in contrast to vanilla diffusion.

## 4.6. Case Study

To illustrate the application of our model, here we provide a case study on topology generation with high stiffness and low density (HSLD). We train our model on a dataset of metamaterials exhibiting better HSLD filtered from our original dataset to emphasize the wanted HSLD attribute. We restrict the density condition to a low value (e.g., 0.3), and we tune the wanted stiffness within a range (e.g., 0.1

to 0.5). Our model can yield topologies as a transiting series along with the transiting properties, as illustrated in Figure 6.

Figure 6 shows that our model suggests octet truss topology in the HSLD-targeted task, while octet truss is known to be a promising candidate for high-stiffness topology (Song et al., 2019). Our model can also generate some novel intermediate topologies that are not in the training dataset, which implies its potential to approximate the intermediate transition within a given distribution and to propose novel metamaterial candidates with wanted properties.

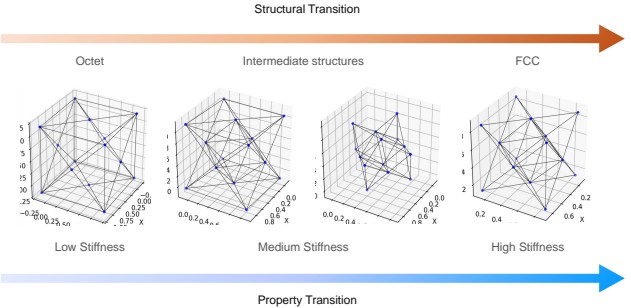

*Figure 6.* Case Study on HSLD metamaterial.

## 5. Related Work

**Material Structure Generation**. Diffusion models (Yang et al., 2023) and variational autoencoders (VAEs)(Kingma, 2013; Chen et al., 2025a) are widely used for material structure generation. SyMat(Luo et al., 2024b), EquiCSP (Lin et al., 2024), and DiffCSP (Jiao et al., 2023) employ diffusion models for crystal topology generation, with SyMat incorporating an additional VAE for lattice vector and atom type generation. These models focus on unconditional generation, lacking property awareness. CDVAE (Xie et al., 2022) integrates a diffusion model with a VAE, encoding structures into a latent space for conditional generation. Cond-CDVAE (Luo et al., 2024a) extends CDVAE with conditional Langevin sampling. For molecular generation, cG-SchNet (Gebauer et al., 2022) and EDM (Hoogeboom et al., 2022) are conditional models: cG-SchNet leverages SchNet (Schütt et al., 2017) for latent space encoding, while EDM processes conditions akin to diffusion time steps. Cond-DFC-VAE (Court et al., 2020) combines crystal generation with property prediction but relies on an external predictor.

**Material Property Prediction**. MACE (Batatia et al., 2022) introduces a high-order message-passing network for graph-based materials, with MACE-ve (Grega et al., 2024) enhancing it through additional components and training modifications. ComFormer (Yan et al., 2024) applies transformers to tokenized crystalline structures. ViSNet (Wang et al., 2024) utilizes graph attention for feature fusion. SphereNet (Liu

et al., 2022) employs 3D spherical representations for message passing. mCGCNN (Meyer et al., 2022) processes embedded node and edge features via CGCNN (Xie & Grossman, 2018). Equiformer (Liao & Smidt, 2022) adapts transformers for graph-based learning with equivariant attributes. SchNet (Schütt et al., 2017) utilizes continuous filters for molecular convolutions, while PaiNN (Schütt et al., 2021) alternates between two types of equivariant message-passing blocks. uniTruss (Zheng et al., 2023a), a VAE-based model for metamaterial property prediction, is not inherently designed for material generation.

# 6. Conclusion

We formalize the diverse mechanical metamaterial design tasks into a general task. Based on the general task, we propose a unified model, UNIMATE, which addresses the data complexity challenge and task diversity challenges. Experiments show that UNIMATE outperforms other baseline models by a significant margin. We also prepare a benchmark to support the training and evaluation of metamaterial design models, which addresses the lack of benchmark challenge. In summary, this paper contributes to problem formalization, model, and benchmark development for mechanical metamaterial design.

# Acknowledgements

We thank the anonymous reviewers for their constructive comments. This work is supported by the National Science Foundation under Award No. IIS-2339989 and No. 2406439, DARPA under contract No. HR00112490370 and No. HR001124S0013, U.S. Department of Homeland Security under Grant Award No. 17STCIN00001-08-00, Amazon-Virginia Tech Initiative for Efficient and Robust Machine Learning, Amazon AWS, Google, Cisco, 4-VA, Commonwealth Cyber Initiative, National Surface Transportation Safety Center for Excellence, and Virginia Tech. The views and conclusions are those of the authors and should not be interpreted as representing the official policies of the funding agencies or the government.

# Impact Statement

This paper presents work whose goal is to advance the field of Machine Learning. There are many potential societal consequences of our work, none which we feel must be specifically highlighted here.

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

# A. Benchmarking Details

## A.1. Dataset

Our data set is built on the basis of the dataset proposed in Lumpe & Stankovic (2021). That dataset contains 17087 topologies, out of which we randomly select 500 topologies with no more than 20 vertices. Figure A.1 illustrates some of the selected topologies.

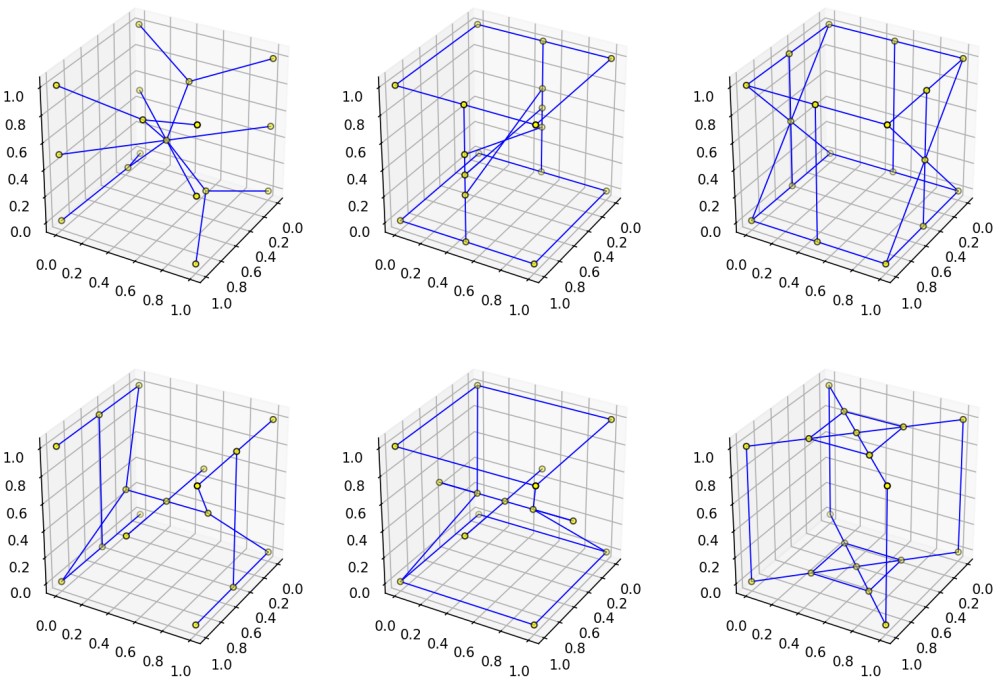

*Figure A.1.* Example Topologies.

Given these 500 topologies, we randomly assign 3 different edge radius for each topology. Given a topology, edge radius and relative density can be deferred based on each other according to:

$$d = \pi r^2 l_{\text{equ}}, \tag{15}$$

where $d$ is relative density, $r$ is edge radius $l_{\text{equ}}$ is the equivalent total edge length:

$$l_{\text{equ}} = \sum_i l_{\text{inner},i} + \frac{1}{2} \sum_i l_{\text{face},i} + \frac{1}{4} \sum_i l_{\text{frame},i}, \tag{16}$$

where the "inner", "face" and "frame" subscripts indicate edges that are within the unit cell, on the surface, or on the outer frame (connecting two near corner vertices). Therefore, edge radius and relative density are equivalent, given a topology. For each topology-density pair, we mesh the 3D structure into small cubic voxels and then apply homogenization simulation (Silveirinha, 2007) to calculate the homogenized mechanical properties of the structure. After such operation, we have 1500 data points (500 topology and 3 densities for each topology, with matching properties). Based on these 1500 data points, we apply data augmentation by rotating the topology and the properties with the same random rotating angle. Each data point is rotated 9 times, so in total, our dataset contains 15000 samples. Our dataset is also provided in the project repository at `https://github.com/wzhan24/UniMate`.

**A.2. Metrics**

We use different metrics for evaluating a model's performance in the three tasks: topology generation, property prediction and condition confirmation.

A.2.1. TOPOLOGY GENERATION

For the generation result assessment, we design two metrics, $F_{\text{qua}}$ and $F_{\text{cond}}$. $F_{\text{qua}}$ is used to measure the quality of a 3D topology, defined as:

$$F_{\text{qua}} = \frac{2F_{\text{sym}}F_{\text{per}}}{F_{\text{sym}} + F_{\text{per}}}, \tag{17}$$

where $F_{\text{sym}}$ measures the central symmetry and $F_{\text{per}}$ measures the periodicity. We denote the vertex coordinates as $X = [\boldsymbol{x}_0, \boldsymbol{x}_1, \ldots, \boldsymbol{x}_q]$, where $\boldsymbol{x} \in \mathbb{R}^3$ is 3D coordinates of a vertex, $q$ is the number of vertices within a unit cell. $F_{\text{sym}}$ is defined as:

$$F_{\text{sym}}(\boldsymbol{X}) = \frac{1}{q} \sum_i \min_j \| \frac{\boldsymbol{x}_i + \boldsymbol{x}_j}{2} - \boldsymbol{c} \|, \tag{18}$$

where $c$ is the centroid coordinates of all the vertices:

$$\boldsymbol{c} = \frac{1}{q} \sum_i \boldsymbol{x}_i. \tag{19}$$

Since all the topologies in our dataset are of cubic shape, we define periodicity metric based on cubic shape. We find the eight corners out of all the vertices by:

$$\boldsymbol{X}_{\text{corner}} = \{\boldsymbol{x}_i | i \text{ in } \mathcal{I}_{\text{corner}}\}, \text{ with } \mathcal{I}_{\text{corner}} = \text{argsort}(-\boldsymbol{X})[0 : 8], \tag{20}$$

$\mathcal{I}_{\text{corner}}$ corresponds to the indices of the eight vertices that are farthest from the centroid. After finding the eight corners, we set them into four pairs, each pair containing one corner and another corner that is farthest from it. Figure A.2 illustrates the four pairs (e.g., corner 1 and 1'). One random corner is set as a anchor corner (e.g., corner 1). For the other 3 pairs (2, 2', 3, 3' and 4, 4'), we consider the one closer to the anchor as a "positive" corner. For example, in the 2-2' pair, corner 2 is closer to corner 1.

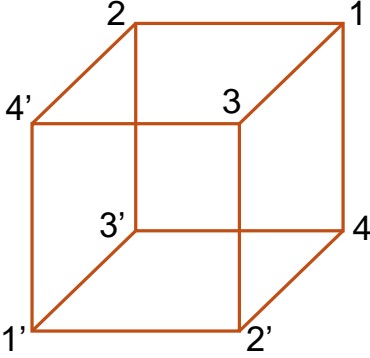

*Figure A.2.* Illustration of Eight Corners.

After matching all eight corners with their relative status (i.e., the topological relation with respect to the anchor), the vector of each outer frame edge can be derived, e.g.:

$$\boldsymbol{e}_{2-1} = \boldsymbol{x}_2 - \boldsymbol{x}_1, \tag{21}$$

where $\boldsymbol{e}_{2-1}$ is the edge vector between corner 2 and corner 1, with other $\boldsymbol{e}_{i-j}$ similarly defined. Then, three average axes can be defined as:

$$\boldsymbol{l}_1 = \frac{1}{4}(\boldsymbol{e}_{2-1} + \boldsymbol{e}_{4'-3} + \boldsymbol{e}_{1'-2'} + \boldsymbol{e}_{3'-4}), \tag{22}$$

$$l_2 = \frac{1}{4}(e_{3-1} + e_{4'-2} + e_{1'-3'} + e_{2'-4}), \tag{23}$$

$$l_3 = \frac{1}{4}(e_{4-1} + e_{3'-2} + e_{2'-3'} + e_{1'-4'}). \tag{24}$$

With the above definition, $F_{\text{per}}$ can be defined as:

$$
\begin{aligned}
F_{\text{per}} = \frac{1}{12}(&\|e_{2-1} - l_1\| + \|e_{4'-3} - l_1\| + \|e_{3'-4} - l_1\| + \|e_{1'-2'} - l_1\| \\
&+ \|e_{3-1} - l_2\| + \|e_{4'-2} - l_2\| + \|e_{2'-4} - l_2\| + \|e_{1'-3'} - l_2\| \\
&+ \|e_{4-1} - l_3\| + \|e_{2'-3} - l_3\| + \|e_{3'-2} - l_3\| + \|e_{1'-4'} - l_3\|).
\end{aligned}
\tag{25}
$$

$F_{\text{cond}}$ is defined as:

$$F_{\text{cond}}(\boldsymbol{X}; \boldsymbol{X_{\text{gt}}}) = \frac{1}{q} \sum_i \min_j \|\boldsymbol{x}_i - \boldsymbol{x}_{\text{gt},j}\|, \tag{26}$$

where $\boldsymbol{X_{\text{gt}}}$ is the ground truth vertex coordinates, with $\boldsymbol{x}_{\text{gt},j}$ being the $j$th ground truth vertex's coordinates. Therefore, $F_{\text{cond}}$ measures how close a generated result is to the ground truth structure that matches the given condition perfectly.

### A.2.2. PROPERTY PREDICTION AND CONDITION CONFIRMATION

For property prediction and condition confirmation tasks, we use NRMSE as the metric, respectively as $\text{NRMSE}_{\text{pp}}$ and $\text{NRMSE}_{\text{cc}}$, defined as:

$$\text{NRMSE}_{\text{pp}} = \frac{1}{\max_{i,j}(p_{\text{gt},i,j}) - \min_{i,j}(p_{\text{gt},i,j})} \sqrt{\frac{1}{N} \sum_{i=1}^N \|\hat{\boldsymbol{p}}_i - \boldsymbol{p}_{\text{gt},i}\|^2}, \tag{27}$$

$$\text{NRMSE}_{\text{cc}} = \frac{1}{\max_i(\rho_{\text{gt},i}) - \min_i(\rho_{\text{gt},i})} \sqrt{\frac{1}{N} \sum_{i=1}^N (\hat{\rho}_i - \rho_{\text{gt},i})^2}, \tag{28}$$

where $\boldsymbol{p}_{\text{gt},i}$ is the $i$th ground truth property (with $p_{\text{gt},i,j}$ being its $j$th component) in the dataset, $\hat{\boldsymbol{p}}_i$ is the output value from the model for the $i$th property, $\rho_{\text{gt},i}$ is the $i$th ground truth density in the dataset, $\hat{\rho}_i$ is the output value from the model for the $i$th density and $N$ is the dataset size.

## B. Experimental Details

### B.1. Implementation Details

We conduct all the experiments on a Linux platform with an NVIDIA A100 GPU (80GB version). We use the same dataset division ratio (i.e.70% for training, 15% for validation and 15% for testing) for each model. We run the training process until each model reaches its best performance, although other models generally require more epochs to converge than our model.

The topology encoder $\mathcal{E}_{\mathcal{T}}$ and decoder $\mathcal{D}_{\mathcal{T}}$ are each a three-layer GCN, with the latent dimension same as the codebook token dimension (e.g., 128). The property encoder $\mathcal{E}_{\boldsymbol{p}}$ and decoder $\mathcal{D}_{\boldsymbol{p}}$, and the density encoder $\mathcal{E}_{\rho}$ and decoder $\mathcal{D}_{\rho}$ are each a three-layer MLP, with the latent dimension same as the codebook token dimension.

### B.2. Algorithm Details

Alg. 1 shows the detailed Tripartite Sinkhorn Algorithm.

## C. Additional Experimental Results

### C.1. Time and Space Efficiency

Table C.1 gives additional results of time and space efficiency analysis results for baselines not mentioned in Section 4.3. Each element in Table C.1 indicates processing time for one batch (seconds per batch) with different size, and "OOM" indicates out of memory error.

---

**Algorithm 1** Tripartite Sinkhorn Algorithm

---

1: **Input:** $\{\tilde{\boldsymbol{X}}^i_{\text{round}}\}_{i=0}^{B-1}, \{\tilde{\boldsymbol{\rho}}^i_{\text{round}}\}_{i=0}^{B-1}, \{\tilde{\boldsymbol{P}}^i_{\text{round}}\}_{i=0}^{B-1}, \epsilon$
2: $F_{\mathcal{T}}, F_\rho, F_{\boldsymbol{p}} = \boldsymbol{0}_n$ //$F$ represents the frequency of tokens, $B$ is batch size, $\epsilon$ is a small positive number
3: $\boldsymbol{u}, \boldsymbol{v}, \boldsymbol{w} = \frac{1}{n}\boldsymbol{1_n}$
4: **for** $i = 0$ **to** $B-1$ **do**
5:     **for** $j = 0$ **to** $l_{\mathcal{T}} - 1$ // $l_{\mathcal{T}} - 1$ is the number of tokens in $\tilde{\boldsymbol{X}}$ **do**
6:         $F_{\mathcal{T},k} \leftarrow F_{\mathcal{T},k} + 1$, if $\boldsymbol{z}_k = \tilde{\boldsymbol{t}}^i_{\text{round},j}$ // $\tilde{\boldsymbol{t}}^i_{\text{round},j}$ is the $j$th token in $\tilde{\boldsymbol{T}}^i_{\text{round}}$
7:     **end for**
8:     Update $F_D$ and $F_P$
9: **end for**
10: Normalize $F_{\mathcal{T}}, F_\rho, F_{\boldsymbol{p}}$
11: $\boldsymbol{C} = \{C_{i,j,k} | C_{i,j,k} = c(z_i, z_j, z_k)\}, \boldsymbol{M} = \exp(-\boldsymbol{C}/\epsilon)$
12: **for** $i = 0$ **to** $N-1$ **do**
13:     $\boldsymbol{u} \leftarrow \frac{F_{\mathcal{T}}}{\boldsymbol{1} \otimes \boldsymbol{v} \otimes \boldsymbol{w} \odot \boldsymbol{M}}, \boldsymbol{v} \leftarrow \frac{F_\rho}{\boldsymbol{u} \otimes \boldsymbol{1} \otimes \boldsymbol{w} \odot \boldsymbol{M}}, \boldsymbol{w} \leftarrow \frac{F_{\boldsymbol{p}}}{\boldsymbol{u} \otimes \boldsymbol{v} \otimes \boldsymbol{1} \odot \boldsymbol{M}}$ //$\odot$ is Hadamard product, $\otimes$ is Kronecker product
14:     **if** $u, v, w$ all converge **then**
15:         break
16:     **end if**
17: **end for**
18: TransPlan $= \boldsymbol{u} \odot \boldsymbol{v} \odot \boldsymbol{w} \otimes \boldsymbol{M}$
19: $d_{\text{w}} = \langle \boldsymbol{C}, \text{TransPlan} \rangle$ //$\langle\,,\,\rangle$ is Frobenius inner product
20: **Return** TransPlan, $d_{\text{w}}$

---

*Table C.1.* Additional Time and Space Analysis Results.

| MODEL\BATCH SIZE | 200 | 1000 | 2000 | 3000 | 5000 | 7000 | 10000 |
|---|---|---|---|---|---|---|---|
| CDVAE (XIE ET AL., 2022) | 0.2416 | 0.4372 | OOM | OOM | OOM | OOM | OOM |
| VISNET (WANG ET AL., 2024) | 0.0285 | 0.0366 | 0.0575 | 0.0772 | 0.1085 | 0.1473 | 0.1931 |
| UNITRUSS (ZHENG ET AL., 2023A) | 0.0108 | 0.01385 | 0.015 | 0.01435 | 0.01406 | 0.01441 | 0.0199 |

## C.2. Parameter Sensitivity

Table C.2 provides additional results for parameter sensitivity analysis. Each element in Table C.2 shows the $F_{\text{cond}}$\NRMSE$_{\text{pp}}$\NRMSE$_{\text{cc}}$ metrics under given latent dimension and number of tokens (i.e., codebook size).

*Table C.2.* Additional Parameter Sensitivity Analysis Results.

| LAT. DIM.\NUM. OF TOKEN | 32 | 64 | 128 |
|---|---|---|---|
| 32 | 0.0836\0.0324\0.0402 | 0.0769\0.0288\0.0436 | 0.0833\0.0294\0.0419 |
| 64 | 0.0779\0.0309\0.0404 | 0.0813\0.0300\0.0409 | 0.0783\0.0295\0.0442 |
| 128 | 0.0795\0.0286\0.0409 | 0.0798\0.0331\0.0430 | 0.0818\0.0282\0.0437 |

