# OpenReview forum: "UniMate: A Unified Model for Mechanical Metamaterial Generation, Property Prediction, and Condition Confirmation"
_ICML.cc/2025/Conference — ICML 2025 poster_

### Official Review · Reviewer_79DH · 2025-03-13

**Overall Recommendation:** 4

**Summary:**

In this work, the authors propose a method that jointly solves topology generation, condition confirmation, and property prediction. Their approach consists of two stages: first, using three sets of encoders and decoders to embed different modality conditions into discrete latents; second, employing a diffusion/score-matching model to generate masked latents from unmasked ones. Experimental results demonstrate significant improvement over the baselines across all three tasks, with ablation studies validating the effectiveness of the proposed modules.

## update after rebuttal
The rebuttal has addressed my minor concerns, and I will maintain my rating.

**Claims And Evidence:**

- This work appears to be the first to propose a unified formulation capable of handling multiple tasks, as shown in Table 1.
- The approach of embedding different modality inputs into a common latent domain for generation is innovative, though similar concepts have been explored recently in image-text applications [1].
- The manuscript's thorough evaluation, benchmarking against existing baselines, and open-source models provide strong evidence for result reproducibility.

[1] Janus: Decoupling Visual Encoding for Unified Multimodal Understanding and Generation

**Essential References Not Discussed:**

Nil.

**Experimental Designs Or Analyses:**

The experiment setting looks reasonable to me, and the experiments show improvement on all tasks over existing baselines. However, since I am not a domain expert for mechanical design, I am not sure if some important baselines are missed in this case.

**Methods And Evaluation Criteria:**

Overall, the proposed method has achieved significant improvement over all baselines on different tasks, including topology generation, property prediction, and condition confirmations. The ablation study has shown the effectiveness of joining encoding of latent spaces and alignment operation.

**Other Comments Or Suggestions:**

Nil.

**Other Strengths And Weaknesses:**

Nil.

**Questions For Authors:**

I have one minor question about the inference process, specifically regarding the computation of the transport plan. For masked tokens, is the marginal distribution initialized directly as a Gaussian distribution? This point could use some clarification in future revisions.

**Relation To Broader Scientific Literature:**

Nil.

**Theoretical Claims:**

As far as I know, this work does not provide a strong theoretical claim on the proposed framework.

---

> ### Author Rebuttal · Authors · 2025-04-01
>
> Thank you for the reviewer’s time and constructive comments. We sincerely appreciate the opportunity to address the concerns and clarify our work.
>
> **Q1:** Similar concepts have been explored in an image-text work (i.e., Janus, which is mentioned by the reviewer).
>
> **A1:** We appreciate the reviewer for pointing out the paper Janus, a multimodal understanding and generation model. While we acknowledge that our work and Janus may share high-level conceptual similarities, there are significant distinctions between the two:
> * **Task Focus**: Janus is designed for general vision-and-language understanding tasks, whereas our model specifically targets metamaterial design, a domain with unique challenges and goals not addressed in Janus.
> * **Methodology**: Janus integrates image information by projecting it into the input space of a large language model (LLM), combining it with textual data for joint understanding. In contrast, our approach projects all three modalities into a shared embedding space that is not biased toward any single modality. This unified representation is better suited to the nature of metamaterial design problems.
>
> **Q2:** The reviewer was not sure whether any important baselines are missed in our work.
>
> **A2:** In our experiments, we have carefully selected representative baselines to provide a fair and comprehensive comparison. To the best of our knowledge, the models we included reflect the current state-of-the-art computer science works in the field of metamaterial design. Still, we are open to including additional baselines should we find any relative suggestions.
>
> **Q3**: For masked tokens, is the marginal distribution initialized directly as a Gaussian distribution?
>
> **A3:** We thank the reviewer for the question regarding token initialization. In our implementation, we initialize tokens based on the transport plan, assigning higher probabilities to tokens with higher values in the transport plan. In response to the reviewer's question, we explore several alternative initialization strategies: pure Gaussian noise, a mixture of transport plan-based initialization, and Gaussian noise. The results of these variants are presented below.
>
> |                   | Fqua   | Fcond  | NRMSE_pp | NRMSE_cc |
> |-------------------|--------|--------|----------|----------|
> | Rand. Noise Init. | 0.0291 | 0.0795 | 0.0304   | 0.0448   |
> | Mixture Init.     | 0.0525 | 0.0814 | 0.0313   | 0.0446   |
> | Trans. Plan Init. | 0.0274 | 0.0781 | 0.0244   | 0.0443   |

---

### Official Review · Reviewer_f6U7 · 2025-03-14

**Overall Recommendation:** 4

**Summary:**

In this paper, the author proposed UNIMAE, a unified model that can tackle three tasks simultaneously, namely, the topology generation task, property prediction task, and condition confirmation task, by training a shared, aligned latent space using a novel TOT and frozen diffusion. In the three tasks, the UNIMAE shows 80.2%, 5.1%, and 50.2% higher performance.

**Claims And Evidence:**

Most of the claims made in this paper are supported by clear evidence.

1.	The author claims that it is a unified model for different tasks in mechanical Metamaterial generation, property prediction, and condition confirmation, which is proven in Table 2

2.	The author claims that the aligned latent space is helpful in handling challenge 1 (mechanical metamaterial design involves three modalities with different formats and distributions.). This is supported in the ablation study case 2

However,
1.	 I noticed that from line 267 to line 269, the author claims that “it will be simpler to generate proper outputs, which increase the robustness of the generation process.”, which is not proven in later experiment

2.	The author claims that using frozen diffusion to handle the challenge 2 (given the complexity of data, any part of the data can be missing and requires design effort. Therefore there can be various design tasks). But no ablation study supports this.

**Essential References Not Discussed:**

N/A

**Experimental Designs Or Analyses:**

Yes, I review all of the experiments.
The main experiment conducted by the author can be summarized as follows:

1.	The main results are shown in Table 2, where different baselines for different tasks are compared with the proposed unified method.

2.	The time and memory efficiency comparison between the proposed method and the baseline models. But not clear why the author only compares three baselines instead of all the baselines mentioned in Table 2.

3.	Choose the latent dimension and the code book size by the Fqua, but it is not clear why choose these two hyperparameters by only one evaluation metric.

4.	Ablation study shows the effectiveness of the latent MTR alignment, but additional ablation study of partially frozen diffusion is needed.

**Methods And Evaluation Criteria:**

The paper proposed a generalized Sinkhorn algorithm by generalizing the transport plan iteration process for the alignment of more than two latent distributions and also uses partially frozen diffusion to tackle potentially different tasks such as mechanical metamaterial generation, property prediction, and condition confirmation with the unified model. These make sense.
As for the evaluation metric, I have some concerns about the validation of the metamaterial generation. It is unclear why Fqua and Fcond are valid metrics for evaluating generation quality. Maybe a FEM simulation is needed to validate the generated metamaterial.

**Other Comments Or Suggestions:**

Further ablation studies are needed such as

1.	the size of the code book and different soft-round strategies: I noticed that in parameter sensitivity, the author provides the Fqua VS. the codebook size and the latent dimension, but what about another evaluation metric? Why determine the hyperparameter with only one single evaluation metric?

2.	The ablation study on partially frozen diffusion is missing.

**Other Strengths And Weaknesses:**

Strengths:

1.	This paper introduces a novel multi-modality alignment method that extends beyond conventional two-modality approaches.

2.	It effectively integrates diffusion-based generation with selective token freezing, enabling flexible handling of multiple tasks.

Weakness:

1.	For topology generation, it is unclear why Fqua and Fcond are valid metrics for evaluating generation quality. A more detailed explanation is needed.

2.	Additionally, the generated topology does not appear to have been validated for effectiveness and manufacturability using numerical simulations. Demonstrating its practical applicability is crucial.

3.	Further ablation studies are needed such as the size of the code book, and different soft-round strategy

**Questions For Authors:**

See weakness

**Relation To Broader Scientific Literature:**

Typically, the feature alignment is done for two modalities. This paper proposes a method to align features from three modalities and is capable of generalizing to more modalities. This can be very useful in other applications.

**Theoretical Claims:**

The main theoretical claims made by the author are

1.	The round operation of latent tokens to the codebook, which is shown from equations 1 to 6.

2.	the proposed generalized Sinkhorn algorithm by generalizing the transport plan iteration process, which is shown from 7 to 9 and algorithm 1 in the appendix.

3.	The partially frozen diffusion, which is shown from equation 10 to 12.

They are all correct.

---

> ### Author Rebuttal · Authors · 2025-04-01
>
> We sincerely thank the reviewer for their time and constructive feedback. Please find our detailed responses and corresponding revisions below:
>
> **Q1:** One point in the work that "it will be simpler...the generation process" is not proven in later experiments.
>
> **A1:** Thank you for your detailed reviews. To address this issue, i.e., the effectiveness of using a codebook for diffusion starting point initialization, we conduct additional experiments (random noise initialization and mixed initialization), with detailed results reported in reply to Q3 from reviewer 79DH.
>
> **Q2:** Ablation for partially frozen diffusion is needed.
>
> **A2:** We appreciate your constructive suggestion. Accordingly, we compare the use of a partially frozen diffusion with a vanilla diffusion setup (i.e., where variables in all dimensions are denoised at each step). The results below verify the effectiveness of partially frozen diffusion. We will include the results in the final version of the manuscript.
>
> |                            | Fqua   | Fcond  | NRMSE_pp | NRMSE_cc |
> |----------------------------|--------|--------|----------|----------|
> | Vanilla Diffusion          | 0.0405 | 0.0813 | 0.0322   | 0.0423   |
> | Partially Frozen Diffusion | 0.0274 | 0.0781 | 0.0244   | 0.0443   |
>
> **Q3:** Further ablation studies are needed, such as the size of the code book and different soft-round strategies.
>
> **A3:** Thanks for the valuable suggestion. According to your suggestion, in addition to the varying codebook sizes experiments conducted in Section 4.4. Below, we provide more results using a soft-round strategy (midpoint interpolation between the initial value and the target rounding token), which shows the effectiveness of codebook rounding.
>
> |                   | Fqua   | Fcond  | NRMSE_pp | NRMSE_cc |
> |-------------------|--------|--------|----------|----------|
> | Soft Round | 0.0421 | 0.0787 | 0.0369   | 0.0449   |
> | Codebook Round | 0.0274 | 0.0781 | 0.0244   | 0.0443   |
>
> **Q4:** It is unclear why Fqua and Fcond are valid metrics for evaluating generation quality; generated topologies are not validated for effectiveness and manufacturability.
>
> **A4:** Thanks for the detailed reviews. Our evaluation metrics are inspired by and adapted from prior literature that adopted similar concepts. Specifically:
> * Fqua evaluates the quality of generated topologies by measuring their symmetry degree and periodicity.
> * Fcond evaluates how well the generated topologies satisfy the input conditions by comparing them with ground truth topologies that fully match those conditions.
>
> Please also refer to Reviewer 44SV **A1** for more detailed illustrations. We will include these detailed illustrations in our final version.
>
> Furthermore, we have collaborated with a metamaterial lab to physically manufacture the generated topologies and test their properties. We plan to release a demonstration video showcasing this process after the double-blind review period concludes.
>
> **Q5:** Why the author only compares three baselines in time and memory efficiency analysis?
>
> **A5:** We appreciate your careful reviews. In the model efficiency analysis section, we present the results for three representative baselines for clarity and brevity. Below, we provide the complete set of results for all baselines, from which we can reach the same conclusion as stated in our manuscript, and we will include these in the appendix of the revised version.
> | Model\Batch Size | 200    | 1000    | 2000   | 3000    | 5000    | 7000    | 10000   |
> |------------------|--------|---------|--------|---------|---------|---------|---------|
> | CDVAE            | 0.2416 | 0.43716 | OOM    | OOM     |       OOM  |   OOM      |       OOM  |
> | VisNet           | 0.0285 | 0.0366  | 0.0575 | 0.0772  | 0.1085  | 0.1473  | 0.1931  |
> | UniTruss         | 0.0108 | 0.01385 | 0.015  | 0.01435 | 0.01406 | 0.01441 | 0.0199  |
>
> **Q6:** In parameter sensitivity, why do the authors only use the metric Fqua?
>
> **A6:** We thank you for the constructive feedback. Accordingly, we include the parameter sensitivity analysis w.r.t. all metrics. In the sensitivity study, since we consider the topology generation task to be relatively hard among the three tasks, and Fqua is a typical indicator of the generation quality, we provide the results related to Fqua. Below, we provide results for the other evaluation metrics as well. These will be added to the appendix in the final version.
>
> Parameter Sensitivity regarding other metrics (Fcond/NRMSE_pp/NRMSE_cc)
> | Lat. Dim\Codebk Size | 32                   | 64                   | 128                  |
> |----------------------|----------------------|----------------------|----------------------|
> | 32                   | 0.0836/0.0324/0.0402 | 0.0769/0.0288/0.0436 | 0.0833/0.0294/0.0419 |
> | 64                   | 0.0779/0.0309/0.0404 | 0.0813/0.0300/0.0409 | 0.0783/0.0295/0.0442 |
> | 128                  | 0.0795/0.0286/0.0409 | 0.0798/0.0331/0.0430 | 0.0818/0.0282/0.0437 |

---

### Official Review · Reviewer_44SV · 2025-03-15

**Overall Recommendation:** 3

**Summary:**

The paper introduces UNIMATE, a unified model for mechanical metamaterial design that simultaneously addresses three key aspects: 3D topology, density condition, and mechanical property. Unlike previous approaches that typically consider only two modalities, UNIMATE integrates all three through a modality alignment module—which compresses and aligns diverse design information into a shared latent space—and a synergetic diffusion generation module that completes missing design tokens via a score-based diffusion model. Experimental results demonstrate that UNIMATE significantly outperforms existing models in topology generation, property prediction, and condition confirmation tasks, offering promising improvements and establishing a new benchmark for comprehensive metamaterial design.

**Claims And Evidence:**

1. The integrated approach is supported by experiments showing notable performance gains ( In the topology generation task, property
prediction task, and condition confirmation task, our model outperforms the second-best model by 80.2%, 5.1%, and
50.2%, respectively), but it relies on a new dataset and custom metrics.

2. The unification and the alignment operation both benefit to the model’s performance. The unification process boosts the performance by 37.5% in average, and the TOT alignment provides 7.8% boost.

**Essential References Not Discussed:**

None

**Experimental Designs Or Analyses:**

1. The authors construct a new dataset (based on Lumpe & Stankovic, 2021) and introduce domain specific metrics (F_qua for topology quality and F_cond for topology matching) to evaluate their model. While these enable comprehensive testing, I would recommend that AC consider having domain experts in material design review its validity

2. They compare UNIMATE against multiple models across three tasks (topology generation, property prediction, condition confirmation). For some tasks, especially condition confirmation, they adapt models not originally designed for that purpose (e.g., forcing property prediction models to predict density). This adaptation might introduce bias, so while the comparisons are informative, they might not fully reflect each baseline’s intended performance.

**Methods And Evaluation Criteria:**

Yes, the quantitive metrics and the ablations are convincing.

**Other Comments Or Suggestions:**

None

**Other Strengths And Weaknesses:**

Strength:
1. Multimodal Alignment: Uses a multimodal codebook to map heterogeneous data (3D topology, density, and mechanical properties) into a unified latent space.
2. Synergetic Generation: Introduces a partially frozen diffusion process with a transformer backbone that generates missing tokens while preserving the context of known tokens.
3. Flexible and Unified Design: Efficiently bridges diverse modalities, enabling robust metamaterial synthesis even with arbitrary missing inputs.
Weakness:
1. One potential weakness of the paper is the limitation in data scale. The approach relies on a newly constructed dataset, which could restrict the model's generalizability and robustness in real-world scenarios.

**Questions For Authors:**

None

**Relation To Broader Scientific Literature:**

None

**Theoretical Claims:**

I believe this is an application paper that employs multimodal alignment and generation for metamaterial synthesis, so it does not propose any specific theoretical novelty.

---

> ### Author Rebuttal · Authors · 2025-04-01
>
> Thanks for the reviewer's time and inspiring comments. Here, we summarize the major points from the reviewer and our rebuttal as follows:
>
> **Q1:** The proposed model is evaluated on a new dataset and custom metrics.
>
> **A1:** Our work focuses on developing a unified model capable of handling diverse metamaterial design tasks as a whole. This is a relatively underexplored area, and to the best of our knowledge, there has not been a comprehensive effort to address it so far. As a result, no suitable dataset currently exists for this task. Therefore, we propose our own dataset tailored to this problem. Additionally, we introduce custom evaluation metrics to assess the quality of generated structures by referring to previous material works. Specifically, the two metrics—Fqua and Fcond—are inspired by previous works [1–4]. References [1,2] emphasize the importance of symmetry and periodicity in metamaterial design, which directly motivates our Fqua metric that quantifies the degree of symmetry and periodicity in the generated topologies. Similarly, references [3,4] propose comparison techniques for assessing topology similarity and generation coverage. Building on these ideas, our Fcond metric evaluates how well the generated structures match the target conditions by comparing their topological features.
>
> *[1]Abu-Mualla, Mohammad, and Jida Huang. "A Dataset Generation Framework for Symmetry-Induced Mechanical Metamaterials." Journal of Mechanical Design 147.4 (2025).*
>
> *[2]Bitzer, Andreas, et al. "Lattice modes mediate radiative coupling in metamaterial arrays." Optics Express 17.24 (2009): 22108-22113.*
>
> *[3]Tian Xie, Xiang Fu, Octavian-Eugen Ganea, Regina Barzilay, and Tommi S. Jaakkola. 2022. Crystal Diffusion Variational Autoencoder for Periodic Material Generation. In ICLR.*
>
> *[4]Nils ER Zimmermann and Anubhav Jain. 2020. Local structure order parameters and site fingerprints for quantification of coordination environment and crystal structure similarity. RSC advances 10, 10 (2020), 6063–6081.*
>
>  **Q2:** The baselines are not originally designed for the tasks in our work.
>
>  **A2:** As mentioned in A1, this is a relatively new task, making it difficult to find baselines that are directly aligned with our objective. This challenge is further amplified by the fact that metamaterial design remains largely underexplored within the computer science community. Nevertheless, we conducted extensive experiments by tuning the hyperparameters of existing baseline methods to better assess their potential performance on our task. Here, we focus on the second best model, uniTruss, as an example. uniTruss has two main hyperparameters, learning rate (LR) and latent dimension. As shown in the following, we report the results on four LR and five latent dimensions. Results show that for both property prediction and density confirmation, the best LR is 5e-4 and the best latent dimension under this LR is 32.
>
> (Property Prediction Task) Tuning learning rate, **bold** denotes the results reported in original manuscript.
> | LR | 1e-3 | **5e-4** | 1e-4 | 5e-5 |
> | -------- | -------- | -------- | -------- | -------- |
> | NRMSE_pp |  0.0285    |   **0.0271**  |  0.0420   |  0.0440   |
> | NRMSE_cc |  0.0899    |   **0.0889**  |  0.106   |  0.115   |
>
> (Property Prediction Task) Tuning latent dimension for property prediction
> | Latent dim | 16 | 32 | 64 | 128 | 256 |
> | ---------- |----|----|----|-----|-----|
> | NRMSE_pp | 0.0328 | **0.0271** | 0.0329 | 0.0298 | 0.0472 |
>
> (Density Confirmation Task) Tuning latent dimension for density confirmation
> | Latent dim | 16 | 32 | 64 | 128 | 256 |
> | ---------- |----|----|----|-----|-----|
> | NRMSE_cc | 0.0905 | **0.0889** | 0.0909 | 0.0899 | 0.891 |
>
>
>
> **Q3:** More experimental details about dataset construction and network design might help readers to reproduce the method.
>
> **A3:** We appreciate the suggestion to include more details about dataset construction and network design. We will incorporate the following information into the appendix to improve the clarity and completeness of the paper.
>
> * For dataset construction, we use homogenization simulation to calculate the property of topologies. We will add details about how homogenization works. Also, we will add details about density selection, e.g., the range from which the edge radius is selected.
> * For network design, we will add more details about each component of our model, including the layer number of the GCN encoder, the MLP encoder, the transformer backbone, the latent dimension of each component, etc. We hope this will help increase the clarity and reproducibility of our work.

---

### Decision · Program_Chairs · 2025-05-01

**Decision:**

Accept (poster)

**Comment:**

The submission introduces a novel method that tackles multiple tasks at the same time: topology generation, property prediction, and condition confirmation.  Reviewers all liked the submission based on its technical contributions and experimental results.  Some concerns about the metrics were also addressed during the rebuttal phase.  The AC would like to follow the recommendation of reviewers and suggest acceptance.